# Seabed Topography Changes in the Sopot Pier Zone in 2010–2018 Influenced by Tombolo Phenomenon

**DOI:** 10.3390/s20216061

**Published:** 2020-10-24

**Authors:** Artur Makar, Cezary Specht, Mariusz Specht, Paweł Dąbrowski, Paweł Burdziakowski, Oktawia Lewicka

**Affiliations:** 1Department of Navigation and Hydrography, Polish Naval Academy, Śmidowicza 69, 81-127 Gdynia, Poland; 2Department of Geodesy and Oceanography, Gdynia Maritime University, Morska 81-87, 81-225 Gdynia, Poland; c.specht@wn.umg.edu.pl (C.S.); p.dabrowski@wn.umg.edu.pl (P.D.); o.lewicka@wn.umg.edu.pl (O.L.); 3Department of Transport and Logistics, Gdynia Maritime University, Morska 81-87, 81-225 Gdynia, Poland; m.specht@wn.umg.edu.pl; 4Department of Geodesy, Faculty of Civil and Environmental Engineering, Gdansk University of Technology, Narutowicza 11-12, 80-233 Gdansk, Poland; pawel.burdziakowski@pg.edu.pl

**Keywords:** tombolo phenomenon, bathymetric surveys, electronic navigational chart, digital sea bottom model

## Abstract

Bathymetric surveys of the same body of water, performed at regular intervals, apart from updating the geospatial information used to create paper and electronic maps, allow for several additional analyses, including an evaluation of geomorphological changes occurring in the coastal zone. This research is particularly important in places where the shape of the coastal zone has been violently disturbed, including by human activity. Tombolo is such a phenomenon and it dynamically shapes the new hydrological conditions of the coastal zone. Apart from natural factors, it may be caused by the construction of hydrotechnical facilities in the littoral zone. It causes a significant disturbance in the balance of the marine environment, resulting in the bottom accretion and dynamic changes in the coastline. This has been the case since 2010 in Sopot, where the rapidly advancing tombolo is not only changing environmental relations but also threatening the health-spa character of the town by stopping the transport of sand along the coast. This paper analyses changes in seabed shape in the pier area in Sopot between 2010 and 2018. In the analysis, both archival maps and bathymetric surveys over a period of 8 years were used; based on these, numerical bottom models were developed and their geospatial changes were analyzed. The results showed that changes in the seabed in this area are progressing very quickly, despite periodic dredging actions organized by administrative bodies.

## 1. Introduction

Bathymetric surveys, performed to obtain geospatial data for the seabed, are undergoing a dynamic evolution both in terms of equipment and measurement methods. Instead of singlebeam echosounders (SBES) [1], multibeam echosounders (MBES) [2] are increasingly used, unmanned survey vehicles (USV) [3,4,5] keep replacing motorboats and hydrographic cutters, and geodetic positioning systems [6] are replacing multi-system Global Navigation Satellite System (GNSS) navigation receivers [7,8,9]. Photogrammetric methods based on UAV [10,11,12,13] and image bathymetry [14,15,16,17] are also increasingly used to analyze shoreline changes [18,19,20,21]. For a decade, laser scanning has become an additional and highly effective measurement method enabling the acquisition of data on the relief and land cover in the coastal strip [22,23]. Please note that the selection of the vessel and its equipment depends on several factors, such as the size of the probed area and distance from the shore, depth, and maneuverability. Therefore, unmanned vehicles [24] can be used with success for measurements in very shallow waters, close to the shore where maneuverability is limited, like in marinas between moored vessels. Their small size allows the use of a singlebeam echosounder, even though multibeam echosounders dedicated to unmanned vehicles are also available. Three major methods of bathymetric measurements are currently available for very shallow waters:Using a typical hydrographic vessel with a very low draught of less than 30 cm (used in the research—Figure 1a,b).Using an unmanned, directly controlled vessel (from the telemanipulator), which means that they are not able to navigate independently on survey profiles.Using an unmanned vessel capable of automatically navigating along a set route.

Another key element in the analysis of geospatial changes in the coastal zone includes modeling of geomorphological processes in coastal line changes [25], including those with an anthropogenic nature [26].

In the literature on the subject, the process of modeling three-dimensional sediment transport has been described in detail and extensively in many publications [27,28]. It includes both the impact on the sea shore, as well as the influence on infrastructure and marine structures [29,30]. The diverse research approach to the authors’ problem results from different methods of modeling taking into account: evolution [31], fixed profile [32], coupling area-line [33], area-crossshore-alongshore transport coupling [27,34], diffusion [30], the wet-dry [35], hybrid [36], line numerical [37] and the cut-cell models [35]. It should be noted with reference to [38] that general wave-average area models may have merits to simulate shoreline evolution for arbitrary geometry. However, the difficulty in treatment of shoreward boundary has hindered the use of this type of approach for simulation of shoreline evolution. The major defect is that it is impossible to simulate morphologic change above the wave-average water level, although it is more or less hidden for macro-tidal environment.

For the sake of navigation safety, marine navigation areas require periodic bathymetric surveys with varied frequency. Navigation areas available for large vessels should be surveyed more frequently than those where only small vessels are operating. This is due to the limited vessel maneuverability, the small reserve under the bottom, and the more serious consequences of a potential collision. When vessels are maneuvering in harbor basins, additional lateral heels occur which reduce the depth reserve under the keel, increasing the demand for reliable bathymetric information in these basins. Based on this, institutions legally responsible for maritime safety determined the frequency of bathymetric surveys both internationally [39] and nationally [40,41]. The lowest measurement frequency has been determined for waters of the exclusive economic zone from the 10-m depth contour or 5 km from the coastline. It is typically every 10–20 years. The highest frequency (every 1–2 years) was established for harbor areas. Special regulations may apply to bathymetric surveys within the framework of inspection of underwater hydrotechnical structures [42]. Here, the greatest frequency (less than one year) was determined for water bodies and offshore extensively operated structures, where the bottom may become significantly shallower. This also applies to maritime structures being a part of passenger, ferry and fuel terminals.

Another area of application for bathymetric surveys includes geomorphology, which studies the relief of the Earth’s surface and the processes creating and transforming them [43,44]. While in both cases it is important to determine the bottom shape, the intended use of this information may vary. Navigation maps provide up-to-date information on depths to guarantee safe navigation or for dredging works. Geomorphological surveys are primarily focused on the assessment of the Earth’s surface relief changing under the influence of natural and anthropogenic processes, changes in morphogenetic processes in the coastal zone against the background of global changes and accelerated rise in ocean levels globally, and the monitoring of the natural environment of selected geo-ecosystems [45].

Morphological changes in the coastal zone and the coastline course are most often due to natural factors [46,47,48]. However, they can also be the result of human activity related to direct interference in the environment, as is the case here, where the marina construction caused a slowdown in the sediment transport along the coast, initiating the process of creating tombolo. The transport of sediments in shallow zones is the main mechanism influencing erosion, beach accumulation, and bathymetry change [49]. The tombolo effect, which occurred near the pier in Sopot (a seaside resort city in Eastern Pomerania on the southern coast of the Baltic Sea in northern Poland, lies between the larger cities of Gdańsk to the southeast and Gdynia to the northwest), is an example of such a phenomenon (Figure 2).

For many years, tombolo has been subject to research from various perspectives: geological, geomorphological, and dynamics of relief-forming processes. It is implemented, among others, in the north-eastern part of the Adriatic Sea [50,51], on the Rhymittyla and Parainen islands and along the Salpausselka III ridge in Finland to study the geology [52], impact on the seashore as well as the impact on marine infrastructure and structures [29,30].

In 2018, the first tombolo survey was carried out for research purposes and it showed significant changes in depth. Moreover, special attention was paid to marine environment protection related to overgrowth. In subsequent studies, the beach was surveyed using laser scanning and unmanned aerial vehicles (UAV) was used to create a digital terrain model (DTM) [53] assessing scan accuracy. Next, the methodology for conducting this type of research was developed [54]. The conclusion was that to obtain a complete geospatial description of the tombolo phenomenon, it is necessary to analyze all archival materials originating from a given area. They would allow for a time–space analysis of the changes. Researchers asked maritime administration bodies for bathymetric archives as these bodies, as part of their statutory activities, carry out periodic bathymetric measurements for Electronic Navigational Chart (ENC) updates and nautical publications. They were re-analyzed to assess geomorphological changes in this phenomenon. The following data were used for tests:Bathymetric surveys from the period before the marina was built. They were performed by the Maritime Office in Gdynia (2010).Bathymetric surveys taken after the marina was built. They were performed by the Maritime Office in Gdynia (2012 and 2015).Bathymetric surveys carried out by us in 2018.Electronic navigational charts made by the Hydrographic Office of the Polish Navy in the years: 2011, 2014 and 2018.

In this paper, for assessment of bottom relief changes, the digital sea bottom model (DSBM) methods known from hydrography were used, described in detail in [55], and were supplemented with an analysis of data reliability [56].

The results are presented in graphic 2D form, while the projection shows the changes in comparison with the measurements from previous campaigns. The paper ends with conclusions, which summarize the most important aspects of the study and set directions for further research.

## 2. Materials and Methods

Due to the very low depth, this area was not always completely surveyed during measurement campaigns. This was due to the size of the vessel and the type of echosounder used. These were SBES and MBES echosounders. Table 1 presents a summary of bathymetric surveys carried out in this area in 2010–2018, information on the echosounder used, and the exact sea area.

To ensure the high reliability in DSBM creation, as many data as possible are needed. This can be achieved by using either MBES or SBES on measuring profiles spaced at small distances of several meters. Therefore, only measurements made in 2010, 2012, and 2015 with a singlebeam echosounder in shallow water proved useful for analyzing changes in bottom relief. Although surveys in 2011 were also taken with a SBES, they do not cover the southern part. Measurements using MBES, taken in 2013, 2015, 2017, and 2018 in deeper water, do not cover the tombolo phenomenon and are thus not useful for analysis. In 2018, the latest surveys in the northern and southern parts using the SBES echosounder were performed.

### 2.1. Archival Hydrographic Data: 2010 and 2012

The first source of geospatial data for analysis was archival materials from the surveys carried out by the Department of Hydrographic Surveys of the Maritime Office in Gdynia. The reporting documentation from bathymetric surveys consisted of hydrographic boards and digital data in the form of Cartesian coordinates and depth related to the chart datum. Graphical information contained on bathymetric boards shows the course of depth contour with spot wise depths and is usually complemented with land infrastructure elements, such as quays, breakwaters, piers. Based on probe arrangement (depth points), the arrangement and course of the measuring profiles, especially in terms of the distance between them, can be derived. It serves as a basis for assessing the value of the material, that was used in the next stage to develop a numerical bottom model. Figure 3 shows two bathymetric charts of the Sopot pier area. The first one comes from 2010, while the second one was created after the marina was completed in 2012.

The chart presented cover area of similar size: in 2010, the surveys were taken on a body of water that measured 450 m × 630 m. Please note that both areas fully cover the area of occurrence of the tombolo phenomenon, which is limited by the marina and the beach and is 600 m long. The surveys under analysis were made with very similar equipment. DGPS receivers (precision of 2 m *p* = 0.95) and a singlebeam echosounder with a depth measurement precision of 1 cm (rms) at 210 kHz were used (Table 2).

### 2.2. Geospatial Data Contained in the Electronic Navigational Charts: 2011, 2014, 2018

The Electronic Navigational Chart (ENC) is the second source of geospatial data for the analysis of bottom relief near the marina in Sopot. Such data are used for the graphic presentation with a System of Electronic Navigational Chart (SENC). It is encoded in the international S-57 standard, containing *geo*, *meta*, *collection* and *cartographic* items. Items of type *geo* contain descriptive characteristics of real-world elements with attributes and acronyms assigned. From the available ENC cells from the studied period, ENC data from the years 2011, 2014, and 2018 were analyzed. Geospatial information contained in the coastline, depth contour and the survey was used to build the DSBM. Depth contours, which are linear items, contain variable horizontal coordinates and a constant vertical component of depth. Figure 4 shows the evolution of the content of the ENC maps over the years. The information was enriched by the marina external breakwater (2014) and the increased surveying density (2018).

Even a rough analysis indicates a change in the coastline course and its shift towards the sea. In addition, waters in the vicinity of the marina became significantly shallower.

### 2.3. Bathymetric Surveys Using SBES: 2018

In 2018, a comprehensive study of the tombolo phenomenon in Sopot began. In December 2018, the first survey of this reservoir was conducted using Gdynia Maritime University Navigator-One, a classic low draught hydrographic vessel. The vessel was equipped, among others, with the Ohmex SonarMite hydrographic singlebeam echosounder and Trimble R10 GNSS receiver with parameters shown in Table 3.

In such measurements, in very shallow water, the survey time gains in importance. Therefore, measurement timing was determined based on a forecast predicting the highest possible water level, using the ecohydrodynamic model developed by the Polish Academy of Sciences, Institute of Oceanology. This enabled the vessel to approach the shore as much as possible, providing depth measurement in very shallow water and maximum data coverage for the body of water. Depth was reduced to chart datum based on data recorded at the mareographic stations of the Institute of Meteorology and Water Management in Gdynia and Gdańsk by interpolating the readings to the measurement area in Sopot. The survey profiles were planned at a distance of 20 m, which coincides with SBES measurements made by the Maritime Office in Gdynia in 2010 and 2012 (10 m).

### 2.4. Comparative Analysis of Research Material

Bathymetric surveys of the water body adjacent to the pier in Sopot made in the years 2010–2018 differ both in terms of implementation technicalities and data processing methods. Furthermore, the data contained in the ENC, SBES and MBES differ not only in numbers but also in geometry, which is related to measurement profile arrangement.

Surveys made using MBES yielded the greatest amount of data. The large number of beams sent from the acoustic transducer in the starboard and port side traverse, the low speed of the survey vessel and high ping rate ensure small distances between the signal reflection points from the bottom, along and across the survey profile. On the contrary, SBES provides a significantly smaller number of bottom points.

To compare such diverse data, DSBM modeling methods known in hydrography were used. In the process of DSBM building, horizontally irregular data were used to create a regular grid (rectangles). Regular data can also be available by exporting from the grd→xyz grid. Both the regular and irregular MBES grid are models with a large scale of data integration. This allows obtaining high reliability of a grid with a small target size when the reliability decreases when building a high-resolution DSBM based on a small set of geospatial data.

The ENC data contain a smaller set of geospatial information that is distributed differently. The depth contours contained in the ENC are parallel to the coastline and the SBES data are perpendicular to the depth contours, which results from the SBES measurement methodology. Additionally, the ENC cells contain geospatial information in SOUNDG objects with a different degree of integration depending on the update (more recent updates contain more data). A singlebeam echosounder usually provides more data than are contained in the ENC and depends on the distance between the profiles. Therefore, while the depth contour waveform is determined precisely on survey profiles, between them it needs interpolation. In principle, it does not have much influence on the depth contour accuracy in an area with low depth changes dynamics.

The use of SBES or MBES results in a different number of measurements made in the same body of water. For comparison purposes, Table 4 presents the number of data measurements included in ENC for the northern and southern water bodies. The southern reservoir, where the tombolo phenomenon occurs, is especially important. Because of the shallow depth, it was not possible to use the MBES echosounder for measurements there. Therefore, the data for the northern body were also presented, where SBES and MBES measurements for ENC cell updates were performed in different years.

The density of survey profiles is another factor influencing the accuracy of DSBM development in this body of water. Figure 5 shows the data coverage of the water body in the vicinity of the pier in Sopot in the years 2010 and 2012, made with the use of the SBES and with distances between the profiles of 10 m. This ensured a dense data coverage as opposed to the surveys taken in 2015 when the distances between the profiles were 90–100 m. In 2018, we tried to keep the distance of 10 m in southern part (tombolo) and 20 m in northern part of the area.

## 3. Results and Discussion

### 3.1. Extraction of Geospatial Data from ENC

Three classes of *geo* objects were used to build the DSBM based on geospatial data contained in the ENC: a survey, which is a set of points of different depths, the coastline, and the depth contour, whose depths are respectively constant. Table 5 presents those elements of the ENC maps that contain geospatial information used to develop DBSM.

These objects are shown in Figure 6. The SOUNDG object is a scattered point object. Data density increased the year ENC map cell was issued. The two remaining objects: COALNE and DEPCNT and linear objects with a large amount of data. Such redundant information is of little use and does not contribute to the increasing reliability of the developed DSBM.

Although the ENC cell is issued with a specific date, the age of the individual data may vary. Figure 7 shows SOUNDG geospatial data of PL5SOPOT cell issued in 2018 with the time span between the oldest and the youngest data being 7 years. For the data marked in red, the year of acquisition is respectively: 2010 (southern part in the beach area—Figure 7a), 2014 (southern part in the corner between the pier and internal breakwater—Figure 7b), 2012 (northern part—Figure 7c) and 2017 (eastern part—Figure 7d).

Non-geospatial data that include information on data acquisition time for two groups of surveys (Figure 7a,b) are the following:

[FE-000092] SOUNDG [PL-0000267704–00001]:1 TG2

Scale minimum: 7999


**Source date: 28 April 2010**


Source indication: PL, PL, reprt, 02000047

Point geometry

3D Edge [VI-0000000025], truncated boundary

[FE-000094] SOUNDG [PL-0000267702–00001]:1 TG2

Scale minimum: 7999


**Source date: 16 June 2014**


Source indication: PL, PL, reprt, 02002316

Point geometry

3D Edge [VI-0000000027], truncated boundary

### 3.2. Digital Sea Bottom Model by Inverse Distance Weighted Method

As with any surface, the description of the seabed bottom is generally a projection of a certain two-dimensional area into space, the coordinates of which are described by means of polynomials of two variables. These can be, for example, Bezier surface patches [57], rational rectangular Bezier patches [58], B-splines, and in Hermite’s polynomials, NURBS (Non—Uniform Rational B—Splines) functions [59] or Coons patches [60]. For building DTM (here: DSBM), the method of a grid of rectangles with the use of interpolation is commonly used. Among the interpolation methods available in the ArcGIS environment, such as Kriging, natural neighborhood and splines, Inverse Distance Weighted (IDW) was used [61,62,63,64]. The value of the *h(x,y)* function at any point (here: in the grid node), is the weighted average of the known depth values from *n* interpolating points and can be presented in the basic form [64]:(1)$$f\left(x,y\right)=\frac{\sum_{i=1}^{n}w_{i}^{\left(k\right)}\left(x,y\right)h}{\sum_{i=1}^{n}w_{i}^{\left(k\right)}\left(x,y\right)},$$
where *n* is the number of interpolating points, *w_i_^(k)^* is the weight of the *i*-th point (the *k* index refers to the type of weight).

The *w(x,y)* weight is a function of distance and determines the magnitude of the influence of the *i*-th point on the interpolated value. Weight coefficients are now mostly calculated according to the relation (1), in which the value of the weight is inversely proportional to the distance between the interpolated point and the measurement point based on the relation [64]:(2)$$w^{\left(k\right)}\left(x,y\right)=\frac{1}{d^{p}},$$
where: *p*—power, $d=\sqrt{{\left(x-x_{i}\right)}^{2}+{\left(y-y_{i}\right)}^{2}}$.

Apart from the vertical component, determined for the node point of the grid, cell size is another parameter selected based on the source points system, resulting from measurements or extracted from ENC cells. Table 6 shows the default values of this parameter for particular bottom models created in the ArcGIS environment. For creating DSBMs, 1 m cell size was set up.

### 3.3. Comparative Analysis of Charts from 2010 and 2012 (SBES)

Based on *xyz* measurement data from the years 2010 and 2012, DSBM was developed in the ArcGIS environment, which is presented in the 2D form in Figure 8. Only one year after the construction of the marina, the depth contour shifting towards the marina could be observed, which is indicative of an increasing shallowing near the beach.

3D imaging is an alternative form of map representation that is close to the natural human perception, hence Figure 9 presents 3D models of the bottom. Figure 9b shows a later model (2012) in grey, which covers the older one (2010) as a result of the shallowing.

In general, waters get shallower throughout the entire presented area, both in the southern and northern parts. This is visible in Figure 9b, where the bottom area, determined based on the 2012 data and marked grey, completely covers the area from 2010. The area with depths above 0.6 m, for which measurement points (Figure 8) are visible, should be considered for analysis. It can be seen in the vicinity of the marina’s southern breakwater that waters became shallower by ca. 1 m.

### 3.4. Comparative Analysis of the 2014 and 2018 (ENC)

For the 2014–2018 period, bottom models were compared based on the data contained in the ENC data obtained with a singlebeam echosounder. The number of data contained in the ENC is greater (Figure 4c and Figure 6) compared to 2011 and 2014 (Figure 4a,b) and is enough for creating the DSBM on the basis of *geo* objects (Figure 10).

Figure 11 shows the shallow areas based on two bottom models created on the basis of the ENC data. The bottom area, determined based on the 2018 ENC data and marked grey, covers the southern area, where tombolo phenomenon takes place.

### 3.5. Comparative Analysis of 2012 and 2018 SBES Soundings

An analysis of the reporting documentation (reporting board with description included in the table describing the boards) of the Maritime Office in Gdynia from the survey carried out in 2012 and our last (2018) survey showed that the measurements were taken down to the draught limit of the survey vessels of 0.6–0.7 m. These measurements do not include shallow water from the 0.6-m depth contour to the coastline. In 2012, distances between the profiles were set at 10 m. For the survey in 2018, the distances were set at 10 m (Sth) to 20 m (Nth). Measurements were taken in a strip of ±400 m wider than the pier. The 2D bottom model based on the SBES surveys in 2018 is shown in Figure 12.

Figure 13 shows the shallow water in 2018 (a) in relation to 2012 (b) in grey. It can be observed in the southern part of the area, between the coastline and the marina.

### 3.6. Cell Size and DBSM Reliability

Cell size is an important parameter in spatial surface modeling (here: DBSM bottom surface) [48,49]. It seems justified to reduce the distance between the grid nodes to obtain a better image. At low measurement density, depths in the nodes are interpolated from distant points, leading to a decrease in reliability of the constructed DBSM. This also affects the calculation of the volume under/over the surface to estimate the gains and losses. Both the volume of the bottom material to be dredged, and the loss, i.e., changes in the bottom shape, are important for the analysis of the tombolo phenomenon.

When measuring with a MBES, the measurement density is high due to the large number of beams sent by the transducer towards the starboard and port side traverse (transverse density). It also depends on the velocity of the survey vessel and pinging frequency (longitudinal distance). The pinging frequency (emission of acoustic impulses by the echosounder) and the transverse distance are affected by depth: as it increases, the transverse distance increases with constant beam separation, and the pinging frequency decreases with increasing MBES operation ranges.

SBES depth measurements allow high resolution geospatial data to be obtained depending on the depth and velocity of the survey vessel, but the traverse distance depends on the distance between measurements on adjacent profiles. When this distance is minimized, the workload increases and is difficult to implement for the helmsman of a larger surveying vessel and under less favorable hydrometeorological conditions (wind, current).

## 4. Conclusions

For the analysis of changes in the relief of the area adjacent to the Sopot pier, especially between the marina and the beach, the available materials with geospatial information were used at the highest possible data density. These were bathymetric measurements taken in 2010, 2012, 2015, and 2018 and the ENC from 2018. This made it possible to build highly reliable numerical bottom models with respect to the actual bottom shape. Because the measurements were made with a singlebeam echosounder on a hydrographic motorboat, the measurements were to the 0.6-m depth contour. The data from the shallow water were interpolated to the shoreline, whose course was obtained based on geodetic field measurements and an Electronic Navigational Chart.

The reliability of the numerical bottom model is influenced by the method and its parameters. For building this numerical bottom model, the IDW method was used without interfering with the parameters such as *power*, *smoothing* or *anisotropy* (*ratio* and *angle*).

Construction of the marina boosted demand for periodic and frequent measurements since the sand began to deposit on the bottom, resulting in water shallowing and widening of the beach, i.e., a shift of the shoreline towards the water. These measurements can be used not only for engineering, i.e., determining how much bottom material has been deposited and thus how many thousands of cubic meters must be removed, but also for studying the sea dynamics. Contemporary bathymetric measuring systems and methods not only make the surveys more accurate but also faster and easier. This paper was written based on geospatial data obtained with a singlebeam echosounder on a hydrographic motorboat, but the use of the same echosounder on the USV enables measurements to be made to a much smaller depth, even 0.2 m. Precise positioning and line keeping of the measuring vessel in automatic mode enables quick measurements on measuring profiles at distances of 2 m and even 1 m. At such shallow depths, it is not necessary to use a multibeam echosounder when the width of the swath is decreasing in the increasingly shallower water. Although the bottom is sandy and hard, it is justified to investigate the possibility of using other echosounder frequencies. The grass and algae that started growing there with the emergence of the tombolo phenomenon affect the interference of the high-frequency echosounder.

By 2012, within just two years, the breakwater area had become shallower by almost 1 m, decreasing its depth from 3.5 to 2.6 m and shifting the 1 m and 1.5 m isobaths by 90 m towards the marina. Continuous transfer of sand resulted in local shallowing, with a depth of 1.5 m, visible on the ENC map from 2018.

Further research on this phenomenon may include a quantitative analysis of the dredged and lost material. The presented research points to where the tombolo phenomenon caused water shallowing. These areas have been marked in grey in the drawings. Although it is possible to calculate the volume of drifted sand, to assess the volume of the dredged material it is necessary to model the target bottom shape after dredging. A decision is needed as to the course of the shoreline and whether the bottom should drop (the depth should increase) evenly, and up to what distance from the shore this should occur. The random character of transport constitutes a limitation in calculating the amount of dredged sand, as it depends on such factors as the diameter of the sediment grain, its weight (taking into account the buoyant force) and broadly defined structural features (packing, sorting, shape) and the roughness of the bottom.

## Figures and Tables

**Figure 1 sensors-20-06061-f001:**
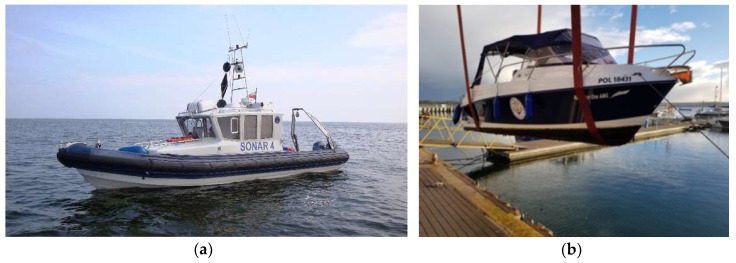
Hydrographic vessels used for surveys in the years of 2010–2015 (**a**) and 2018 [24] (**b**).

**Figure 2 sensors-20-06061-f002:**
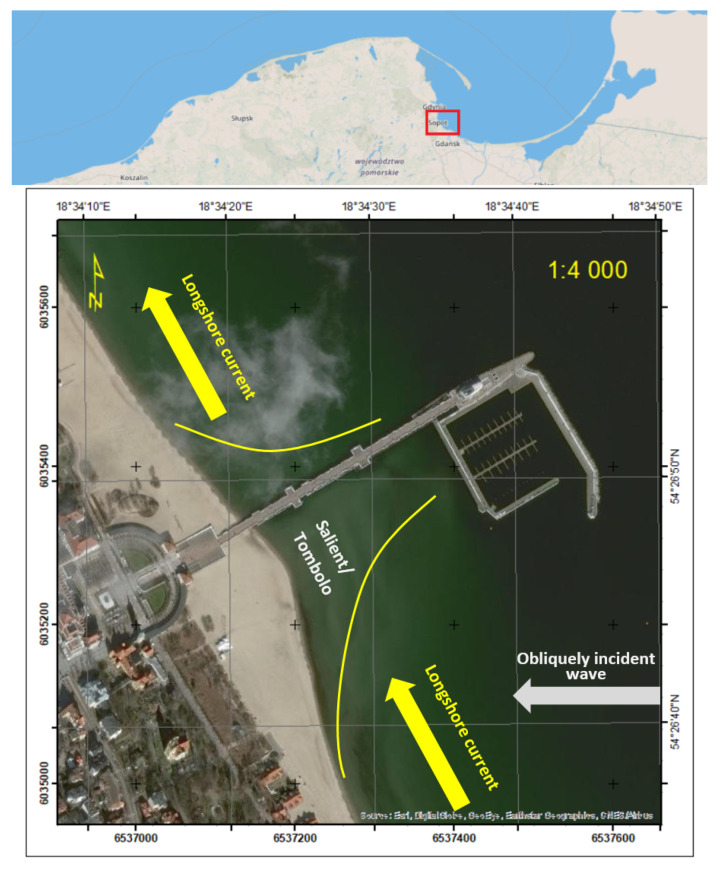
The nature of the phenomenon that causes the formation of the tombolo effect [24].

**Figure 3 sensors-20-06061-f003:**
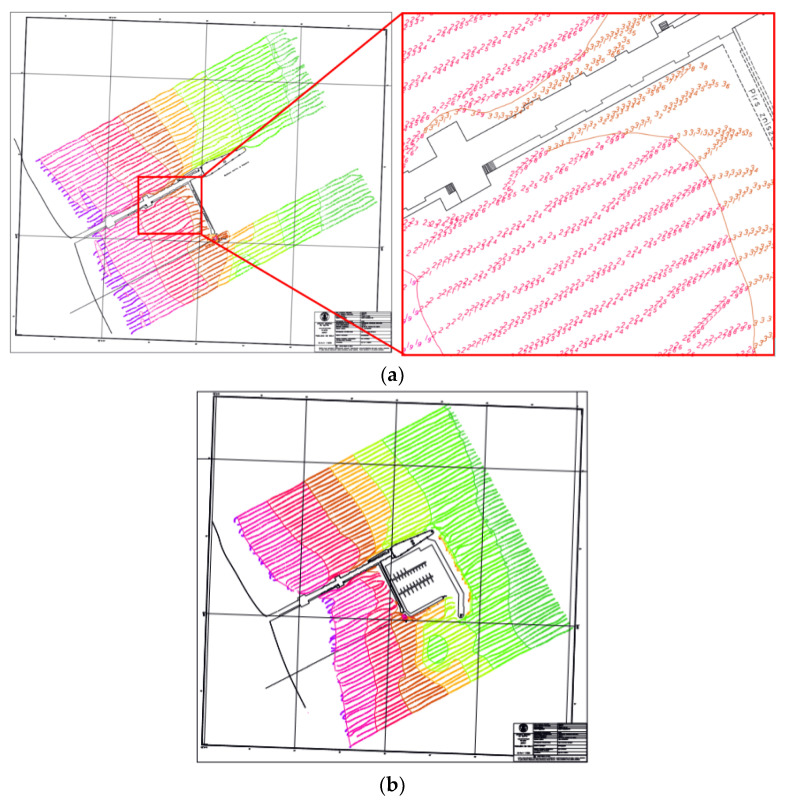
Bathymetric charts showing the variability in bottom relief in 2010 (**a**) and 2012 (**b**). North oriented in the scale of 1:1000

**Figure 4 sensors-20-06061-f004:**
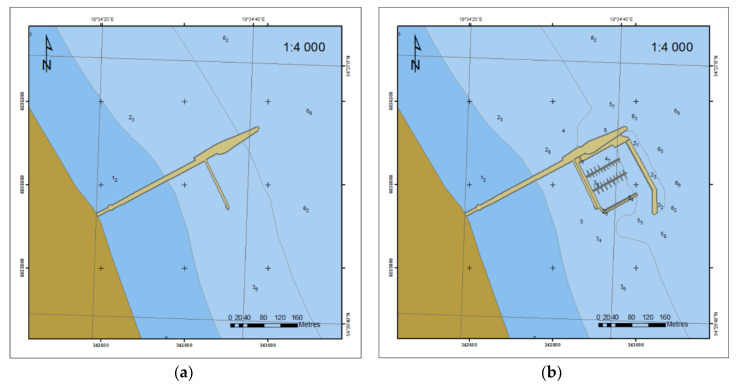
Evolution of Electronic Navigational Chart (ENC) content in 2011 (**a**), 2014 (**b**) and 2018 (**c**).

**Figure 5 sensors-20-06061-f005:**
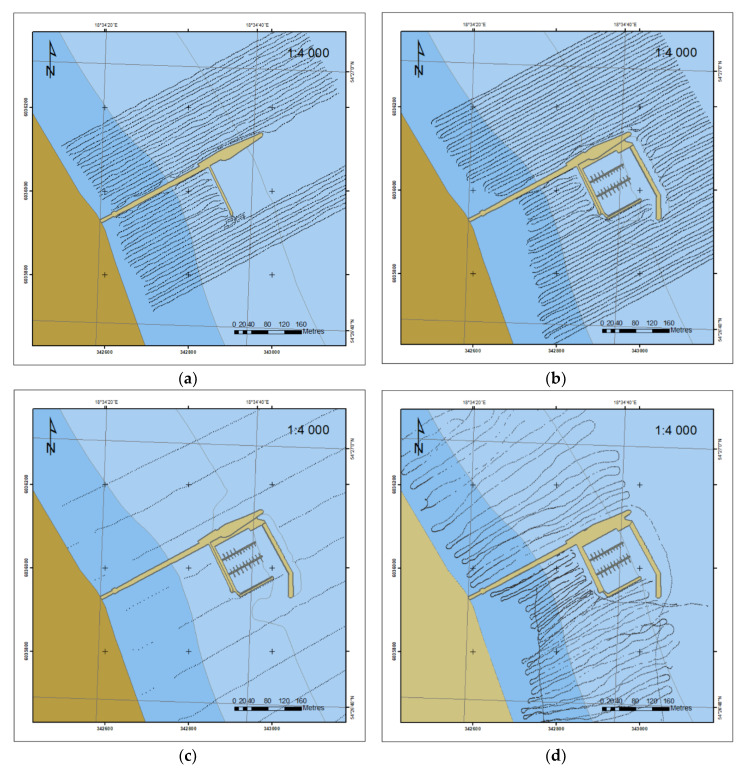
Data coverage with SBES survey in 2010 (**a**), 2012 (**b**) 2015 (**c**) and 2018 (**d**).

**Figure 6 sensors-20-06061-f006:**
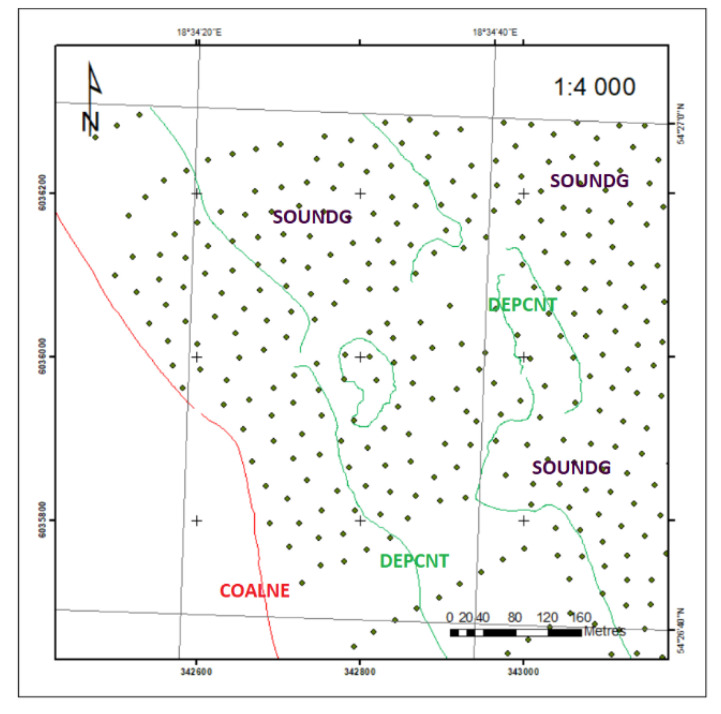
Geospatial data extracted from ENC in 2018.

**Figure 7 sensors-20-06061-f007:**
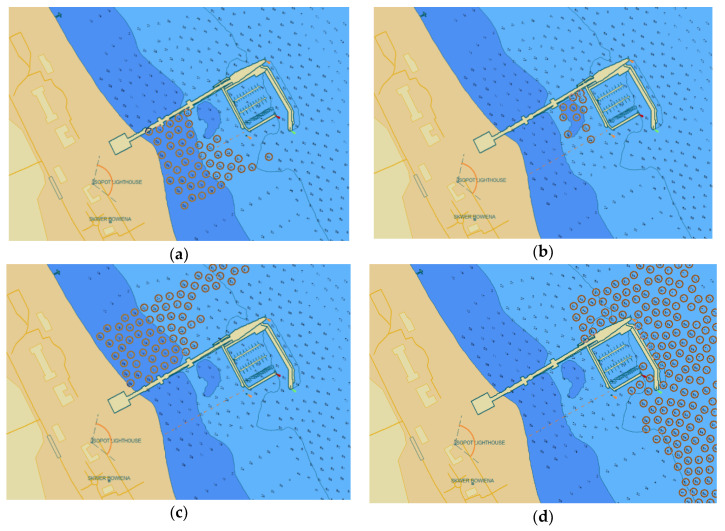
Data acquisition time: 28 April 2010 (**a**), 16 June 2014 (**b**) 22 June 2011 (**c**), 01 March 2017 (**d**).

**Figure 8 sensors-20-06061-f008:**
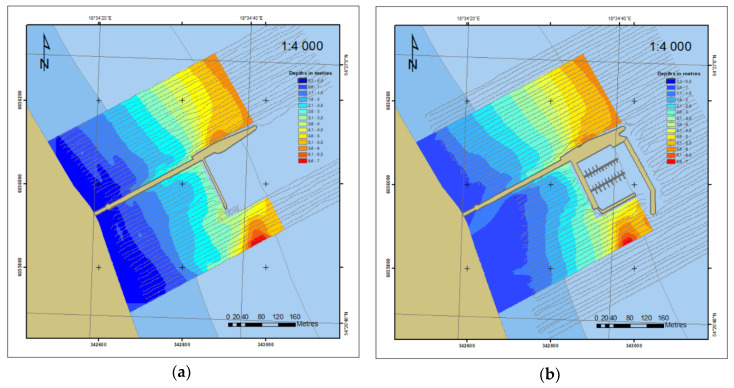
2D bottom model based on the 2010 (**a**) and 2012 (**b**) surveys.

**Figure 9 sensors-20-06061-f009:**
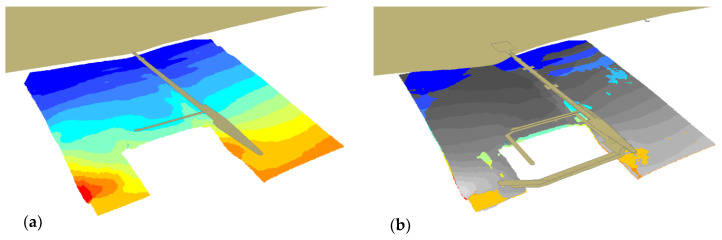
3D bottom model based on the 2010 (**a**), 2010 and 2012 surveys (**b**).

**Figure 10 sensors-20-06061-f010:**
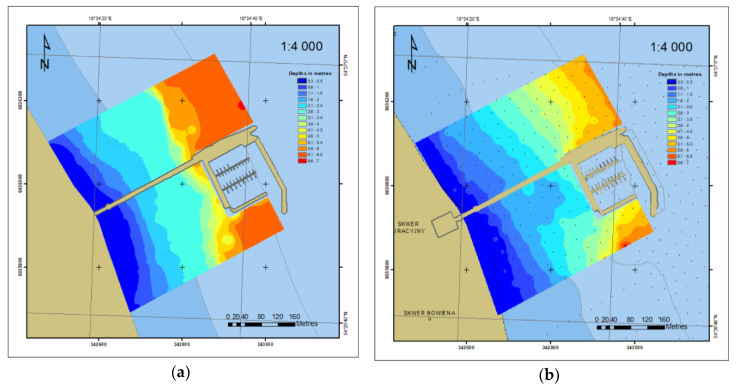
2D bottom model based on the 2014 (**a**) and 2018 (**b**) ENC data.

**Figure 11 sensors-20-06061-f011:**
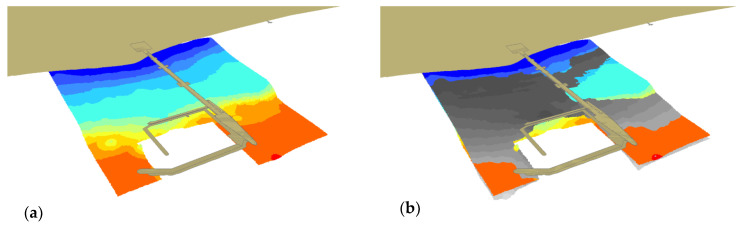
3D bottom model based on the 2014 (**a**) and 2018 (**b**) ENC data.

**Figure 12 sensors-20-06061-f012:**
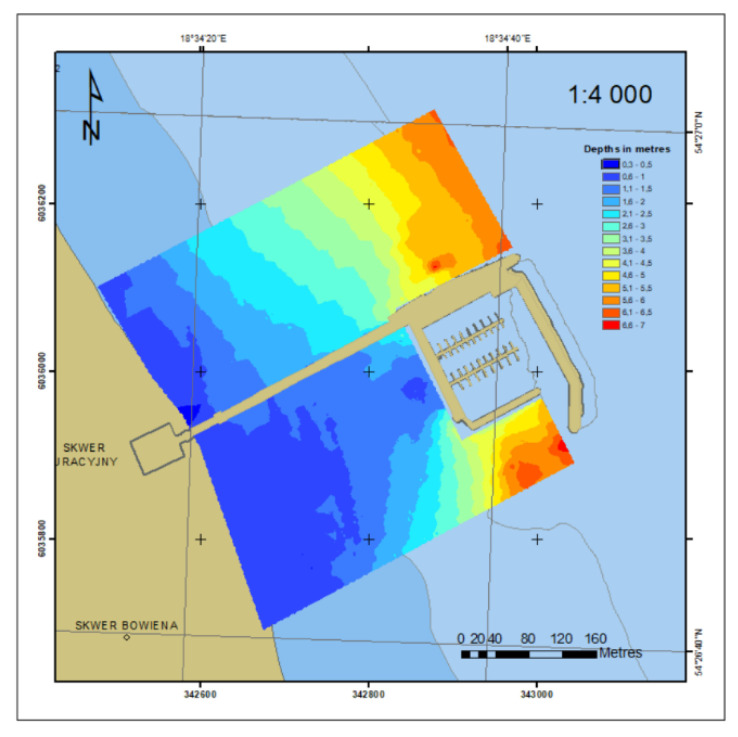
2D bottom model based on the SBES surveys in 2018.

**Figure 13 sensors-20-06061-f013:**
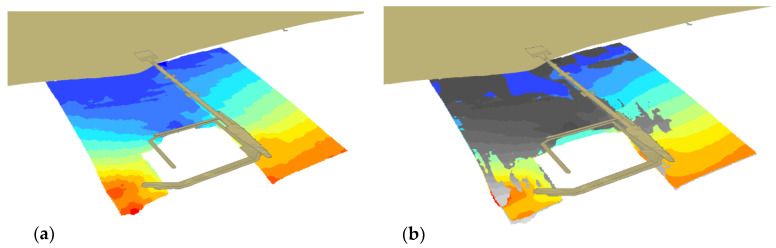
3D bottom model based on the 2018 (**a**), 2012 and 2018 (**b**) SBES surveys.

**Table 1 sensors-20-06061-t001:** Measurements in the pier area in Sopot and the bathymetric equipment used.

Year	Echosounder	Model	North ^1^	South ^2^	East ^3^
2010	SBES	Reson Navisound 515	+	+	
2011	SBES	Reson Navisound 515	+		
2012	SBES	Reson Navisound 640	+	+	
2013	MBES	Reson Seabat 8125			+
2015 01	MBES	Reson Seabat 8125			+
2015 05	SBES	Reson Navisound 640	+	+	+
2017	MBES	Kongsberg EM3002			+
2018	MBES	R2Sonic 2024			+
2018	SBES	Ohmex SonarMite	+	+	

^1^ The northern part, north-west of the pier, ^2^ the southern part, south-east of the pier between the beach and the inner marina breakwater, ^3^ the eastern part, east of the outer marina breakwater.

**Table 2 sensors-20-06061-t002:** Major measurement parameters around the pier in Sopot in 2010 and 2012 campaigns.

Date	28 April 2010	28 June 2012
Positioning	Trimble NT300D	Trimble SPS 461
Depth measurement	Reson Navisonud 515	Reson Navisonud 640
Projection	UTM	UTM
Reference system	WGS84	WGS84
Chart datum	500 NN_55_ Harbor Master’s Office—Northern Port in Gdańsk	500 NN_55_ Harbor Master’s Office—Northern Port in Gdańsk

**Table 3 sensors-20-06061-t003:** Basic parameters of the elements of the hydrographic system.

Device	Parameter	Value
SBES Ohmex SonarMite	Frequency	200 kHz
	Beam spread	8°–10°
	Depth range	0.3–75 m
GNSS Receiver Trimble R10	Systems	GPS + GLONASS
	Frequency	L1 + L2
	GBAS	VRS.net

**Table 4 sensors-20-06061-t004:** Number of data contained in ENC and SBES.

Area	Year	Data Source	Total Number of Data	Number of Data Used
Southern	2010	SBES	4066	2875
	2011	ENC	41	13
	2012	SBES	7312	2415
	2015	SBES	2305	127
	2015	ENC	298	87
	2018	SBES	6520	3436
	2018	ENC	249	101
Northern	2010	SBES	4999	3313
	2011	ENC	76	30
	2012	SBES	6163	3425
	2015	SBES	845	178
	2015	ENC	130	83
	2018	SBES	3788	2042
	2018	ENC	194	144

**Table 5 sensors-20-06061-t005:** ENC elements containing geospatial information used to develop DBSM.

Object	Acronym	Type	Code
Coastline	COALNE	L	30
Depth Contour	DEPCNT	L	43
Sounding	SOUNDG	P	129

**Table 6 sensors-20-06061-t006:** Default value of cell size.

Year	Data Source	Cell Size (m)
2010	SBES	2.778
2011	ENC	3.838
2012	SBES	3.237
2014	ENC	3.838
2015 05	SBES	6.428
2018	SBES	3.639

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
