# Peer review of "Seabed Topography Changes in the Sopot Pier Zone in 2010–2018 Influenced by Tombolo Phenomenon"

_sensors, 2020, doi:10.3390/s20216061_

Round 1
Reviewer 1 Report
Dear author,
i think that the new version of the paper is improved with respect to the previous one. The paper is generally well-organized and the conclusions are supported by presented data. A few minor comments are highlighted in the annotated manuscript.
Even if I'm not a mother tongue, i would suggest a review of the English because some parts of the manuscript are still wordy and not completely clear.

Author Response
Dear Reviewer,
Thank you for your comments. Below is a response to comments.
Line 24: “uplift” has been replaced by “accretion”
Line 29, 59, 79, 89, 123, 392, 396, 399, 405 have been corrected
Line 83, 176, 177: words are in red – they are corrected on the basis of reviewer’s suggestion
Line 256: geo is in italic as one of four types of feature object in S-57 standard
Line 418-420: a sentence has been removed
Thank you for these observations.
Best regards,
Artur Makar

Reviewer 2 Report
Manuscript ID: sensors-968924-peer-review-v1
Revisions requested
I read with interest the notes that the author has included in his review and I greatly appreciated his effort in the editing of the new text.
This is the second revision of the text "Seabed Topography Changes in the Sopot Pier Zone in 2010-2018 Influenced by Tombolo phenomenon" that I read with interest. I consider the subject of the text of particular importance for the interactions between sedimentary processes and changes in natural balances due to maritime works.
As I wrote in the previous review, The paper aims to monitor over time the evolution of the seabed in correspondence of a Marina, with cartographic methods. This is described in the abstract in which the morphological and sedimentary dynamics aspects are cartographed by the processing of numerical data and their modelling in the creation of digital models of the seabed.
The definition of Tombolo, already in the title and in the enunciation at the first place in the keywords, denotes the importance to be given to the sedimentological phenomenon. I must, however, note that both in the introduction chapter and in the text and in the discussion, the geomorphological dynamics of the coast and the seabed are not discussed, but are only taken as a measure of the seabed.
Below are the general notes to the paragraphs of the draft:
Introduction:- The introduction lacks a paragraph with the geographical and geomorphological classification of the area. In fact, the coast in study is very interesting not only for the sedimentary implications due to the construction of the marina and the reconstruction of the pier in 2010, but also for the location of the coast in a wide bay whose arching is consequent to the action of the wave fronts and currents that develop. In fact, on the west side of the bay of Zatoka there is a large and elongate Spit accumulated by the transport of sediments (littoral drift) to the South-East. This phenomenon combined with the wave motion, determines the rotation of the gulf coastline, at least the westernmost part. The wave motion, in this stretch of coast on which the pier and the marina are built, generates a persistent flow along the coast of sediments towards North-West (sedimentary drift), which could be responsible for the asymmetry of the sedimentary salient, more developed South-East of the pier.
The authors in Figure 2 (not interpreted in the new one) and in the text not discussed the sedimentary dynamics. In the short note of Metrosea 2019, Genoa, "Metrological aspects of the Tombolo effect investigation - Polish case study", the same authors briefly describe the evolution of that stretch of coast. The authors have also recently presented the evolution of the area, in the paper (ref.54) and in a more recent: “Mariusz Specht, Cezary Specht, Oktawia Lewicka, Artur Makar, Paweł Burdziakowski and Paweł Dąbrowski. 2020. Study on the Coastline Evolution in Sopot (2008–2018) Based on Landsat Satellite Imagery. Journal of Marine Science and Engineering 8(6):464. DOI: 10.3390/jmse8060464” in which the temporal evolution of the salient at the Pier is indicated. The other geomorphological indications are in my previous review.
I think that for the authors it is possible to shortly summarize the geomorphological characteristics of the area and indicate the references. This integration of the text would put the reader in a position to understand the morpho-dynamic context in which the salient grows.
Row 109, Figure 2: Inappropriate caption. The image does not explain the nature of the phenomenon and the growth of sedimentary salient.
Row 175 Salient accumulation for tombolo phenomenon
Figure 6: You would delete insets (a) and (b) because the data density is poor and they are only re-processed ENC charts. These two insets are useful for viewing coastline changes, but these are already shown in Figures 4 and 5. I suggest that fig. 6 shows only the inset (c), enlarged and with better readability, with added objects: COALINE, SOUNDING, DEPCTN.
I propose to explain better (or change): rows 270 to row 272 including figure7. Part to be deleted or rewritten. The authors do not explain why an overall map of partial reliefs 7 years apart was created. Beyond the numerical treatment of non-coeval data, it must be underlined that the processes of sedimentary dynamics linked to the construction of maritime works are sometimes very fast and interdependent behaviors.
In general: in some figures complete legend, scale, North and captions and the dimension of the characters.
Final Remarks: the paper is interesting and the available data are very abundant. I believe that even with the changes suggested, the draft does not lose its importance and scientific value.
I have verified that the authors made many required changes, improving the text.
I believe, in any case, that a brief geographical and geomorphological overview of the area and the discussion of some sedimentary processes responsible of the sea bottom modifications and the creation of the "salient" (which were stated in the previous version of the text), should be taken into account, rather than delete them from the text.
I encourage the authors to produce a good revision of the text, which promises excellent results.
Author Response
Dear Reviewer,
Thank you for your comments. Below is a response to comments.
Introduction:- The introduction lacks a paragraph with the geographical and geomorphological classification of the area. In fact, the coast in study is very interesting not only for the sedimentary implications due to the construction of the marina and the reconstruction of the pier in 2010, but also for the location of the coast in a wide bay whose arching is consequent to the action of the wave fronts and currents that develop. In fact, on the west side of the bay of Zatoka there is a large and elongate Spit accumulated by the transport of sediments (littoral drift) to the South-East. This phenomenon combined with the wave motion, determines the rotation of the gulf coastline, at least the westernmost part. The wave motion, in this stretch of coast on which the pier and the marina are built, generates a persistent flow along the coast of sediments towards North-West (sedimentary drift), which could be responsible for the asymmetry of the sedimentary salient, more developed South-East of the pier.
A short information about location Sopot city has been added (in bracket). Sediment transportation has been presented in Fig. 2 in graphic form
The authors in Figure 2 (not interpreted in the new one) and in the text not discussed the sedimentary dynamics. In the short note of Metrosea 2019, Genoa, "Metrological aspects of the Tombolo effect investigation - Polish case study", the same authors briefly describe the evolution of that stretch of coast. The authors have also recently presented the evolution of the area, in the paper (ref.54) and in a more recent: “Mariusz Specht, Cezary Specht, Oktawia Lewicka, Artur Makar, Paweł Burdziakowski and Paweł Dąbrowski. 2020. Study on the Coastline Evolution in Sopot (2008–2018) Based on Landsat Satellite Imagery. Journal of Marine Science and Engineering 8(6):464. DOI: 10.3390/jmse8060464” in which the temporal evolution of the salient at the Pier is indicated. The other geomorphological indications are in my previous review.
Fig. 2 has been corrected – extended with the sedimentary dynamics
I think that for the authors it is possible to shortly summarize the geomorphological characteristics of the area and indicate the references. This integration of the text would put the reader in a position to understand the morpho-dynamic context in which the salient grows.
Row 109, Figure 2: Inappropriate caption. The image does not explain the nature of the phenomenon and the growth of sedimentary salient.
I think, now the caption is appropriate after correction the figure
Row 175 Salient accumulation for tombolo phenomenon
Figure 6: You would delete insets (a) and (b) because the data density is poor and they are only re-processed ENC charts. These two insets are useful for viewing coastline changes, but these are already shown in Figures 4 and 5. I suggest that fig. 6 shows only the inset (c), enlarged and with better readability, with added objects: COALINE, SOUNDING, DEPCTN.
Insets (a) and (b) have been deleted and objects have been added to the ENC dated 2018
I propose to explain better (or change): rows 270 to row 272 including figure7. Part to be deleted or rewritten. The authors do not explain why an overall map of partial reliefs 7 years apart was created. Beyond the numerical treatment of non-coeval data, it must be underlined that the processes of sedimentary dynamics linked to the construction of maritime works are sometimes very fast and interdependent behaviors.
This part presents data acquisition time of selected soundings in one ENC dated 2018. In the area of interest, we have old data: 2010 (close to the beach) and 2014 (close to the marina)
In general: in some figures complete legend, scale, North and captions and the dimension of the characters.
Final Remarks: the paper is interesting and the available data are very abundant. I believe that even with the changes suggested, the draft does not lose its importance and scientific value.
I have verified that the authors made many required changes, improving the text.
I believe, in any case, that a brief geographical and geomorphological overview of the area and the discussion of some sedimentary processes responsible of the sea bottom modifications and the creation of the "salient" (which were stated in the previous version of the text), should be taken into account, rather than delete them from the text.
Thank you for these observations.
Best regards,
Artur Makar

This manuscript is a resubmission of an earlier submission. The following is a list of the peer review reports and author responses from that submission.
Round 1
Reviewer 1 Report
This manuscript employed both historical data and a recent bathymetric survey to investigate seabed changes in the vicinity of a pier. Based on my reading neither the sensors employed were not developed as part of this study nor a rigorous uncertainty analysis of the different data bases was conducted. Furthermore, observation regarding the morphological changes were not correlated with wave conditions in this area during the period of study. Thus, the results are limited to a qualitative comparison between bathymetric surveys. I believed this not contribute to improve the state of the art. Hence, I think that significant work is required so the manuscript is worth to be published in a high-impact journal.
Reviewer 2 Report
The paper "Seabed Topography Changes in the Sopot Pier Zone
in 2010-2018 Influenced by Tombolo Phenomenon" of Artur Makar et al is a study dealing with the comparison of available single-beam bathymetric surveys to study the progressive formation of a Tombolo coastward of the Sopot Marina. The paper is quite well-organized, except for some parts (see below) and the conclusions are generally supported by results.
In my opinion, the paper is mostly addressed to show results from single-beam surveys, so i don't understand the need of parts associated to multibeam system in "data and methods" and "results". Differently, in the introduction it is fine to deal with multibeam systems, because they are commonly used for this kind of activities, even if i agree with the authors that both methods are comparable at very shallow water sectors. On the contrary, the authors should better evidence the part relative to the morphological changes that occurred in the area, showing for instance difference maps obtained as comparison between successive surveys and describing the areal and volumetric computation through a more quantitative way (even if they were estimated by DTM with different reliability).
The paragraph 3.5 Comparative analysis of ENC (2018) chart and SBES (2018) soundings can be moved in data and methods.
The figures should be improved as suggested in the annotated manuscript, because scale or color bar are missing or other figures can be merged in a single one or moved in supplementary material. Several detailed comments on the text are also present in the annotated file, highlighting unclear sentences or repetitions.

Reviewer 3 Report
Overall this is a very nice manuscript. I have only a few minor edits and comments.
General comments:
- Overall, the English grammar is quite good. There are a few rough spots, but the journal editors should take care of these during proof.
Specific comments:
- Line 72. I think that the reference to [37] should be [34], “… with reference to [37]…” should be “…with reference to [34]…”. Or maybe the second citation [34] should be [37]?
- Line 87. Replace “largest” with “highest”.
- Line 182. Should “accuracy” be “precision”? It seems that the authors are specifying the precision of the measurement to within 1 cm.
- Figure 4. State the unit of bathymetry measurements (I’m assuming they are meters, but US charts are in feet).
- Lines 208-2016. Providing the amplitude of the tide would be useful here.